# Evolution of Shape and Volume Fraction of Superconducting Domains with Temperature and Anion Disorder in (TMTSF)$_2$ClO$_4$

**Kaushal K. Kesharpu** [1], **Vladislav D. Kochev** [1] and **Pavel D. Grigoriev** [1,2,3,*]

1 Department of Theoretical Physics and Quantum Technology, National University of Science and Technology "MISiS", 119049 Moscow, Russia; kesharpu.k@misis.ru (K.K.K.); vd.kochev@misis.ru (V.D.K.)
2 L.D. Landau Institute for Theoretical Physics, 142432 Chernogolovka, Russia
3 P.N. Lebedev Physical Institute of RAS, 119991 Moscow, Russia
* Correspondence: grigorev@itp.ac.ru

**Abstract:** In highly anisotropic organic superconductor (TMTSF)$_2$ClO$_4$, superconducting (SC) phase coexists with metallic and spin-density wave phases in the form of domains. Using the Maxwell-Garnett approximation (MGA), we calculate the volume ratio and estimate the shape of these embedded SC domains from resistivity data at various temperature and anion disorder, controlled by the cooling rate or annealing time of (TMTSF)$_2$ClO$_4$ samples. We found that the variation of cooling rate and of annealing time affect differently the shape of SC domains. In all cases the SC domains have oblate shape, being the shortest along the interlayer $z$-axis. This contradicts the widely assumed filamentary superconductivity along the $z$-axis, used to explain the anisotropic superconductivity onset. We show that anisotropic resistivity drop at the SC onset can be described by the analytical MGA theory with anisotropic background resistance, while the anisotropic $T_c$ can be explained by considering a finite size and flat shape of the samples. Due to a flat/needle sample shape, the probability of percolation via SC domains is the highest along the shortest sample dimension ($z$-axis), and the lowest along the sample length ($x$-axis). Our theory can be applied to other heterogeneous superconductors, where the size $d$ of SC domains is much larger than the SC coherence length $\xi$, e.g., cuprates, iron-based or organic superconductors. It is also applicable when the spin/charge-density wave domains are embedded inside a metallic background, or vice versa.

**Keywords:** organic superconductors; Beechgard salts; Maxwell-Garnett approximation; high-Tc

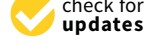



## 1. Introduction

Raising the superconducting transition temperature (Tc) has been the goal of active research for a century. Compounds like cuprates [1,2], iron-based superconductors [3], organic superconductors (hereafter denoted as OrS) [4] are some major high-Tc superconductors at ambient pressure. These materials have several common properties: (i) layered crystal structure and, hence, high conductivity anisotropy; (ii) interplay between various types of electron ordering, i.e., between spin/charge-density wave and superconductivity; (iii) spatial inhomogeneity. Therefore, many effects and methods, both experimental and theoretical, are common for these materials. A recent and good review of all these systems is given by Stewart [5]. Our paper concerns the OrS and is devoted to two problems: (i) show that Maxwell-Garnett approximation can be used to estimate superconducting volume fraction in coexistence regime of superconducting, metallic and spin/charge-density wave phases; (ii) analysis of corresponding experimental data [6,7] in the organic superconductor (TMTSF)$_2$ClO$_4$ to study the effect of cooling rate and of disorder on the volume fraction, shape and size of SC domains.

(TMTSF)$_2$X series belongs to quasi-1D OrS and has been widely studied for 40 years [4,8–10]. Many important effects have been discovered and investigated on these compounds,

e.g., angular magnetoresistance oscillations (AMRO) in quasi-1D metals [9–11], field-induced spin-density waves (FISDW) [12,13], etc. [9,10]. An interesting and puzzling property of these materials related to our subject is that with the increase of pressure for $(TMTSF)_2PF_6$ or of anion ordering for $(TMTSF)_2ClO_4$, the superconductivity first appears along the least-conducting $z$-axis, while along the most conducting $x$-direction only in the last turn [6,7,14,15]. Recent experimental study on $(TMTSF)_2PF_6$ [14–17] and $(TMTSF)_2ClO_4$ [6,7] has shown that spin-density wave (SDW), superconducting (SC) and metallic phase coexist in form of segregated domains. With the increase in pressure (for $(TMTSF)_2PF_6$) [14,15,18] or in anion ordering (for $(TMTSF)_2ClO_4$) [6,7] the volume fraction of SC or metallic phase increases.

A few theories have been suggested to describe the coexistence regime in these materials, especially in $(TMTSF)_2PF_6$. One of them is the application of SO(4) symmetry that explains SDW and SC coexistence [19] but does not account for the observed hysteresis [16], for the strong enhancement of the upper critical field $H_{c2}$ [20] and for the anisotropic SC onset [14,15,18] in the coexistence phase. Less exotic theories suggest a separation of SDW and SC in a coordinate [14–16,20–24] or momentum space [23]. The momentum SC-SDW separation assumes a semi-metallic state in a SDW phase, where, due to the imperfect nesting, small ungapped Fermi-surface pockets appear and become superconducting [23]. The $H_{c2}$ enhancement can be explained in both scenarios [23,24], but the observed hysteresis suggests a spatial SDW/SC separation[16]. To explain the $H_{c2}$ enhancement [20] the SC domain width must not exceed the in-plane SC penetration depth $\lambda_{ab} \equiv \lambda$. In $(TMTSF)_2ClO_4$ the in-plane penetration depth is [25–27] $\lambda_{ab}(T=0) \approx 1$ μm, and the out-of-plane penetration depth is [28] $\lambda_{bc}(T=0.19 \text{ K}) \approx 40$ μm. One possible mechanism of the formation of such narrow domains in the SDW state could be the soliton phase [14,21,22,24,29]. It suggests that SDW order parameter becomes non-uniform with metallic domain appearing perpendicular to the highest conducting $x$-axis. However, the width of these soliton-wall domains, being of the order of SDW coherence length $\xi_{SDW} \sim 30$ nm, is too small to be consistent with the recent observation of AMRO and FISDW in $(TMTSF)_2PF_6$ [15] and in $(TMTSF)_2ClO_4$ [30], suggesting the domain size $d > 1$ μm. Moreover, the soliton-phase scenario accounts for the SC suppression along the most conducting $x$-axis only, but it could not explain why SC first appears along the least-conducting $z$-axis, because in this scenario the soliton walls are extended along both $y$ and $z$-axes, which should result in SC along both these directions.

A probable explanation of this anisotropic SC onset was proposed recently [31]. It is based on two ideas. First, in anisotropic media the isolated SC islands increase conductivity much stronger along the least-conducting direction than along the others, as observed in FeSe [32,33] and described [32–34] using the Maxwell-Garnett approximation (MGA) [35] for small volume fraction $\phi$ of SC phase. However, this MGA theory cannot explain the anisotropic zero-resistance onset. For this we need the second idea, which takes into account the finite sample size $L$ as compared to the size $d$ of SC grains. If the sample shape is very anisotropic, e.g., a thin plate, then the current percolation via the SC grains, responsible for the zero-resistance onset, is most probable along the shortest sample dimension, [31] i.e., along the sample thickness, and least probable along the sample length. These two ideas were applied to explain [31] the experimental data[14,15,18] in PF6, taken on thin elongated samples with dimensions [14,18] $3 \times 0.2 \times 0.1$ mm$^3$. As $(TMTSF)_2ClO_4$ samples are usually flat shaped either [6,7], similar ideas can be applied to analyze the resistivity experimental data there too. In this paper, using the MGA we find superconducting volume ratio $\phi$ and shape of inclusions from available experimental data. We also investigate how disorder and cooling rate affect the inclusions' shape and size. This knowledge may help to better understand the microscopic structure and electronic properties of the SDW/SC coexistence phase in organic superconductors.

In Section 2 we briefly describe the important properties of $(TMTSF)_2ClO_4$ and argue that MGA can be applied to estimate temperature dependence of SC volume ratio ($\phi$) in the presence of all 3 phases, i.e., metallic, SDW and SC. In Section 3 we describe the theoretical

model in MGA. Furthermore, in Section 4 we apply our theoretical model to analyze the experimental data on $(TMTSF)_2ClO_4$. Using the experimental data from Ref. [7] we find out how cooling rate effects $\phi$. Similarly, using the experiments of Ref. [6] we analyze the evolution of aspect ratio of SC inclusions with sample disorder. Finally, in Section 5 we discuss the main results of our investigation and their consequences.

## 2. Material and Method

### 2.1. Material

$(TMTSF)_2ClO_4$ is the only member of Beechgard salts which becomes superconducting at ambient pressure [36,37]. It is a quasi-1D superconductor in which cooper-pair ordering (SC) coexists with insulating Pierels ordering (SDW) (The analysis below equally applies for a charge-density wave (CDW) or SDW Pierels ordering. Our current study is mainly devoted to SDW/SC coexistence in $(TMTSF)_2ClO_4$, therefore we keep SDW notation for the insulating phase. However, it may equally be applied to other compounds with CDW/SC mixed phase.). This very competition between SC and SDW is the key to understand the unconventional superconductivity [38–41]. Among quasi-1D superconductors $(TMTSF)_2ClO_4$ is the only compound in which superconductivity can be controlled by the cooling rate of samples, which affects the disorderliness of $ClO_4$ anions. Slowly cooled $(TMTSF)_2ClO_4$ samples (relaxed state) undergo an SC transition at $T_c \approx 1.3$ K [42,43]. However, when cooled very fast (quenched state) $(TMTSF)_2ClO_4$ has an insulating SDW transition at $T_{SDW} \approx 4$–5 K [44–46]. This behavior can be ascribed to structural change that occurs at the anion-ordering temperature [47] $T_{AO} \approx 24.5$ K. For $T > T_{AO}$ the noncentrosymmetric tetrahedral $ClO_4$ anions, which are located at the inversion centers [48,49], preserve the inversion symmetry due to thermal motion of $ClO_4$ anions. $ClO_4$ anions randomly occupies one or other orientations, hence, on average the inversion symmetry is preserved [48]. For $T < T_{AO}$, if the sample is cooled fast enough then the randomness of orientation of $ClO_4$ anions is preserved [45]. However, if the sample is cooled slowly through $T_{AO}$, $ClO_4$ anions along *a*,*c*-axes are ordered uniformly; and along *b*-axis ordered alternatively [47,50,51]. The anion ordering introduces the new wave vector and the Fermi-surface folding. It disturbs the Fermi-surface nesting, preventing the SDW and favoring SC.

Recent experiments strongly support the presence of SC inclusions embedded in the background of metallic/SDW phase when $(TMTSF)_2ClO_4$ is cooled at intermediate rate [6,7]. This means granular superconductivity for partially ordered samples. From the crystallographic point of view, in this coexistence phase the domains of ordered $ClO_4$ anions are embedded inside disordered background. In these domains the anions have alternating ordering pattern, being disordered outside the domains [6,7,47]. The electronic state inside these domains is assumed to remain metallic even at $T < T_{AO}$, and at $T < T^*$ these $ClO_4$-ordered grains become superconducting [7]. Here $T^*$ is the superconducting onset temperature. The volume fraction $\phi$ of superconducting phase increases with the decrease of temperature. For slow or intermediate cooling rate at $T = T_c$ the phase coherence of these SC islands establishes in the entire sample, leading to its zero resistance. In the temperature interval $T_c < T < T^*$ the effective medium model for highly anisotropic heterogeneous layered compounds [35], developed and applied by the authors to various anisotropic compounds in Refs. [32–34,52], can be used to estimate the volume fraction and the shape of SC inclusions inside the samples.

### 2.2. Method

To analyze the temperature dependence of resistivity at $T_c < T < T^*$ we use the Maxwell-Garnett approximation (MGA) [35], valid when the volume fraction of SC phase $\phi \ll 1$. Please note that the condition $\phi \ll 1$ is fulfilled in a wide range of parameters, because in 3D anisotropic samples even the SC percolation threshold $\phi_c$, corresponding to the onset of nearly zero resistance, is considerably smaller than unity. This is illustrated in our recent work [31], where $\phi_c$ in needle-shaped flat $(TMTSF)_2PF_6$ samples from Refs. [14,18], typical for organic superconductors, was calculated numerically, assuming a rectangular

sample shape and ellipsoidal SC inclusions of varying size at randomly distributed but fixed positions. This calculation has shown the strong anisotropy and small value of these $\phi_c$ (see Figures 3 and 4 of Ref. [31]). Thus, for SC domain size $d = 40$ μm the percolation threshold along the $x$-axis $\phi_c \approx 0.3$; for $d = 15$ μm $\phi_c \approx 0.15$ (see Figure 3 of Ref. [31]). For y- and z-axes $\phi_c$ is even smaller. In all these cases $\phi_c \ll 1$, suggesting a wide range where MGA can be applied. The $(TMTSF)_2ClO_4$ samples, studied in the present paper and in Refs. [6,7], are also flat and needle-shaped, similar to $(TMTSF)_2PF_6$ samples from Refs. [14,18].

In recent works [32–34,52] using MGA the superconducting volume ratio $\phi$ was found when SC inclusions were embedded inside metallic background. Similarly, here we also assume SC inclusions are embedded inside a background phase. However, here the background phase consists of SDW and metallic phases. This is permissible in MGA approximation as long as we know the effective conductivity of this mixed background phase. We make two more assumptions: (i) the SC inclusions are of ellipsoidal shape, which simplifies the calculations and allows deriving analytical formulas for conductivity; (ii) the size $d$ and distance between SC inclusions $l$ are much greater than the SC coherence length $\xi$, so that the SC proximity effects and the Josephson coupling between the SC grains do not change the results considerably. Both in $(TMTSF)_2PF_6$ and $(TMTSF)_2ClO_4$ the size $d \gtrsim 1$ μm of metal/SC inclusions is indeed much larger than the SC coherence length [7,53] $\xi = 70, 30$, and 2 nm along the $a, b$, and $c$ axes, respectively. Such a large metal/SC grain size is demonstrated by the observation [15,30] of angular magnetoresistance oscillations (AMRO) in the mixed phase of these compounds. If $\phi \ll 1$, one can also take $l \gg \xi$.

Due to the proximity effect a shell of SC condensates around SC inclusions with thickness $\sim \xi$ gets created, which changes the effective size and shape of SC islands. Another quantum mechanical effect is the Josephson coupling between SC grains. The Josephson coupling energy $E_J$ depends directly on Josephson junction current $I_c$, $E_J \equiv \hbar I_c/2e$, which exponentially decreases with the increase of distance $l$ between neighboring SC grains: $I_c = I_0 \exp(-l/\xi)$. If $E_J \ll T$, the Josephson coupling can be disregarded [54]. As we assume $l, d \gg \xi$, both these effects can be neglected.

The conductance through N-S boundaries of a normal metal and SC may increase, maximum two times, due to the Andreev reflection (See Section 11.5.1 of Ref. [54] for the basic description of Andreev reflection). However, this increase of the interface conductance should not considerably change the effective conductivity of the whole sample because the main voltage drop (at a given current) comes not from the N-S interfaces, but from the large non-SC parts of the sample. Even if we take an infinite N-S interface conductance, this does not much affect the total sample resistance.

Considering the above facts, we neglect these small quantum mechanical effects, and treat conductivity due to embedded SC inclusions using the classical MGA approximation and percolation.

## 3. Theory

We use the Maxwell-Garnett approximation (MGA) [55,56] to find the SC volume ratio $\phi$. We derive a general formula for effective conductivity in anisotropic heterogeneous media, where unidirectional ellipsoidal SC (SDW) inclusions are embedded inside the background SDW/metallic (metallic) phase with anisotropic resistivity. We denote the diagonal effective conductivity tensor of this materials as $diag(\sigma_{xx}, \sigma_{yy}, \sigma_{zz})$. Similarly, the background conductivity tensor of this material is denoted as $diag(\sigma_{xx}^b, \sigma_{yy}^b, \sigma_{zz}^b)$. Background phase consists of both metallic and SDW phase, or only of the metallic phase. Since the MGA in its standard form[35] can be applied only to an isotropic medium, we use the coordinate-space dilation to transform the initial problem with anisotropic conductivity into isotropic one [32–34] (see Appendix A for the details of this mapping and Equations (A4) and (A5) for the definition of dilation coefficients $\mu$ and $\eta$).

Before explaining the idea of MGA [35,55,56] we note that the stationary-current equation for electrostatic potential $V(r)$, coming from the continuity equation for the electric

current $J_i$ in heterogeneous media with coordinate-dependent diagonal conductivity $\sigma(\mathbf{r})$, (In Equations (1) and (2) the summation is assume over the repeated coordinate indices $i$ and $j$.)

$$-\nabla_i J_i = \nabla_i\big[\sigma_{ij}(\mathbf{r})\nabla_j V(\mathbf{r})\big] = 0, \tag{1}$$

is completely equivalent to the electrostatic equation in heterogeneous media with coordinate-dependent dielectric constant $\varepsilon(\mathbf{r})$, as comes from the Maxwell's equations:

$$\nabla_i D_i \equiv \nabla_i(E_j \varepsilon_{ij}) = -\nabla_i\big[\varepsilon_{ij}(\mathbf{r})\nabla_j V(\mathbf{r})\big] = 0, \tag{2}$$

where $E_i$ and $D_i$ are the electric and displacement fields, and $\nabla_i \equiv \partial/\partial x_i$. Hence, the problem of an effective dielectric constant of such a medium with heterogeneous dielectric function $\varepsilon(\mathbf{r})$ is equivalent to the problem of effective conductivity of a heterogeneous medium with the same coordinate dependence of $\sigma(\mathbf{r})$.

To explain the idea of MGA (Originally, MGA was formulated in 1904 only for spherical inclusions in isotropic medium [57].) [35,55,56] of calculating the effective dielectric constant of heterogeneous isotropic media with a small volume fraction $\phi$ of inclusions of the second phase, we refer to Figure 1. The background phase is represented as "2", and the inclusion phase as "1" in the figure. We take a sphere of our heterogeneous media. Inside this sphere the inclusions are embedded as shown in Figure 1. Let this sphere be embedded in an infinite medium of background phase. An electric field $E$ is applied to the infinite medium. Let $\mathcal{E}'$ be the electric field at a distant point P. $\mathcal{E}'$ includes polarization due to individual inclusions. We denote the inclusion volume ratio as $\phi$ and the background-phase volume ratio as $(1-\phi)$. We assume that the heterogeneous sphere with both inclusion and background phase can be substituted by a sphere of homogeneous phase with an effective dielectric constant $\varepsilon_{eff}$. Then the electric field at point P can be found in two ways: (i) By summing the polarization effect due to individual inclusions; the corresponding electric field at point P we denote as $\mathcal{E}'$. (ii) By taking the polarization effect at point P of homogeneous sphere with effective dielectric constant $\varepsilon_{eff}$; the corresponding electric field at point P we denote as $\mathcal{E}''$. The MGA assumes these fields to be the same, i.e., $\mathcal{E}' = \mathcal{E}''$, which gives an equation on $\varepsilon_{eff}$ (See Section 18.1.1 of Ref. [35] for complete discussion on MGA). Using the well-known formula for the polarization of dielectric ellipsoid in isotropic medium, and replacing $\varepsilon_{ii}$ by $\sigma_{ii}$, we find the following equation (see Equations (18.9) and (18.10) of Ref. [35]) for the effective conductivity $\sigma_i^*$ along main axis $i$ in the mapped space: (Equations (3), (5) and (8) assume that in the mapped space the conductivities of both phases, $\sigma^{isl}$ and $\sigma^b$, are isotropic. For a generalization to anisotropic $\sigma^{isl}$ or $\sigma^b$ in the mapped space see Refs. [58,59]. In the case of superconducting inclusions, applied in Section IV to analyze the experimental data, $\sigma^{isl} = \infty$ is naturally isotropic.)

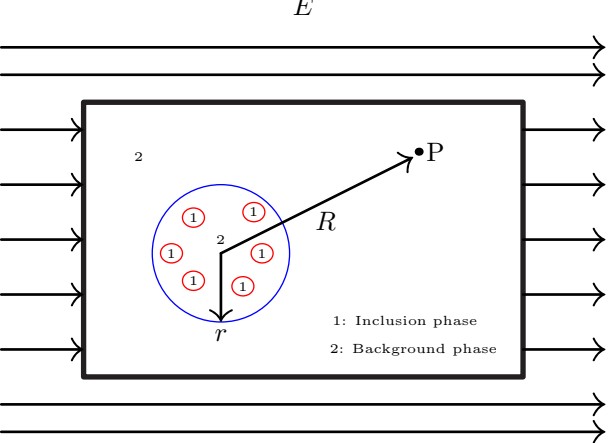

**Figure 1.** Schematic representation of Maxwell-Garnett approximation. The distance of point P from the center of sphere is very large compared to the sphere size, i.e., $r \ll R$. $E$ is the applied electric field.

$$(1-\phi)(\sigma_i^* - \sigma^b) + \phi \left[ \frac{\sigma^b\left(\sigma_i^* - \sigma^{isl}\right)}{\sigma^b + A_i\left(\sigma^{isl} - \sigma^b\right)} \right] = 0. \tag{3}$$

Here $A_i$ are the depolarization factors of a dielectric ellipsoid with semiaxes $a_i$ in the mapped space:

$$A_i = \prod_{n=1}^{3} a_n \int_0^\infty dt \left/ 2(t+a_i^2)\sqrt{\prod_{n=1}^{3}(t+a_n^2)}\right. . \tag{4}$$

The analytical solution of this integral can be found in terms of incomplete elliptic integrals of first and second kind (See Appendix B of Ref. [34]). If the inclusions have finite conductivity, then solving Equation (3) for $\sigma_i^*$ we obtain

$$\sigma_i^* = \sigma_b \left[ \frac{(A_i + (1-A_i)\phi)(\sigma^{isl} - \sigma^b) + \sigma^b}{A_i(1-\phi)(\sigma^{isl} - \sigma^b) + \sigma^b} \right]. \tag{5}$$

If the inclusions are superconducting, we take $\sigma^{isl} \approx \infty$. Equation (3) in this case simplifies to

$$(1-\phi)(\sigma_i^* - \sigma_i^b) - \phi\frac{\sigma^b}{A_i} = 0 \tag{6}$$

Solving Equation (6) for $\sigma_i^*$ we obtain

$$\sigma_i^* = \sigma^b \left[ \frac{A_i + (1-A_i)\phi}{A_i(1-\phi)} \right] \tag{7}$$

Both Equations (5) and (7) gives the effective conductivity $\sigma^*$ in the mapped space. Although the background conductivity in the mapped space is isotropic, the effective conductivity $\sigma^*$ is anisotropic because of the anisotropic ellipsoidal shape of inclusions. The effective conductivity of original anisotropic material in real space is found by the reverse mapping. It is done via multiplying the effective conductivity matrix in the mapped space $\sigma^* = diag(\sigma_{xx}^*, \sigma_{yy}^*, \sigma_{zz}^*)$ by the inverse mapping coefficient matrix $diag(1, \mu, \eta)$. Hence, multiplying Equation (5) by $diag(1, \mu, \eta)$, we find the effective conductivity in real space:

$$\begin{aligned}
\frac{\sigma_{xx}}{\sigma_{xx}^b} &\equiv \frac{\sigma_{xx}}{\sigma^b} = \frac{(A_x + (1-A_x)\phi)(\sigma^{isl} - \sigma^b) + \sigma^b}{A_x(1-\phi)(\sigma^{isl} - \sigma^b) + \sigma^b}, \\
\frac{\sigma_{yy}}{\sigma_{yy}^b} &\equiv \frac{\sigma_{yy}}{\mu\sigma^b} = \frac{(A_y + (1-A_y)\phi)(\sigma^{isl} - \sigma^b) + \sigma^b}{A_y(1-\phi)(\sigma^{isl} - \sigma^b) + \sigma^b}, \\
\frac{\sigma_{zz}}{\sigma_{zz}^b} &\equiv \frac{\sigma_{zz}}{\eta\sigma^b} = \frac{(A_z + (1-A_z)\phi)(\sigma^{isl} - \sigma^b) + \sigma^b}{A_z(1-\phi)(\sigma^{isl} - \sigma^b) + \sigma^b}.
\end{aligned} \tag{8}$$

Here $\sigma_{ii}^b$ is the background conductivity along the axis $i$ in real space. Similarly, multiplying Equation (7) by $diag(1, \mu, \eta)$, we find the effective conductivity of original inhomogeneous material with SC inclusions of volume fraction $\phi$:

$$\frac{\sigma_{ii}}{\sigma_{ii}^b} = \frac{A_i + (1-A_i)\phi}{A_i(1-\phi)}. \tag{9}$$

From Equation (9) one can express the volume fraction $\phi$ via the effective $\sigma_{ii}$ and background $\sigma_{ii}^b$ conductivities and depolarization factor $A_i$ along the same axis:

$$\phi = \frac{A_i(1 - \sigma_{ii}^b/\sigma_{ii})}{A_i + (1-A_i)\sigma_{ii}^b/\sigma_{ii}}. \tag{10}$$

Equation (8) is helpful when the conductivities of background and inclusion phases are both finite. Hence, it can be used to find the effective conductivity of heterogeneous material, when, e.g., SDW domains are embedded inside a metallic background, or vice versa. Equation (9) can be used when superconducting inclusions are embedded inside a background phase of finite conductivity. Below we use Equations (9) and (10) to analyze experimental resistivity data in (TMTSF)$_2$ClO$_4$ in the mixed SC/SDW state. Here the background phase is made up of metallic as well as SDW phases. Resistivity along the corresponding axis is found by taking the inverse of Equations (8) and (9).

## 4. Analysis of Experimental Data in (TMTSF)$_2$ClO$_4$

We consider partially ordered (TMTSF)$_2$ClO$_4$ samples. We denote $T^*$ as superconducting onset temperature. In these compounds for $T > T^*$ there is no superconductivity. However, for $T < T^*$ the domains containing ordered ClO$_4$ anions partially transform to superconducting inclusions [7]. These ordered domains are embedded inside the phase of unordered ClO$_4$ anions, where SDW prevails but may coexist with metallic phase. Further cooling results in the increase of SC volume fraction $\phi$ and in the formation of coherent clusters of SC inclusions. At $T = T_c < T^*$ a complete SC channel gets opened [6,7], i.e., the SC phase coherence establishes in the whole sample. In Section 4.1, we use our theory to calculate the SC volume ratio $\phi$. In Section 4.2, we study the influence of cooling rate on SC volume ratio. In the end, in Section 4.3, we find the approximate shape of SC inclusions in various disordered samples.

### 4.1. Application of MGA Theory to Describe Resistivity and to Find Superconducting Volume Ratio $\phi$

To find a typical temperature dependence of SC volume ratio $\phi(T)$ in (TMTSF)$_2$ClO$_4$ at cooling rate $-dT/dt \leq 100$ K/min we choose the sample #4 in Figure 2 of Ref. [6] (This sample #4 was cooled at rate $-dT/dt = 100$ K/min and then annealed for some time at varying temperature between 15 and 23 K. Therefore, its disorder, presumably, corresponds to a slower cooling rate.) due to the availability of experimental resistivity data on $\rho_{xx}(T, H = 2T)$ in a magnetic field $H = H_z$, shown in Figure 2c of Ref. [6]. The magnetic field destroys superconductivity, and we can use these data to find the conductivity of background phase $\sigma_{xx}^b(T) = 1/(\rho_{xx}(T, H = 2T) - \Delta\rho_{xx})$ (see the inset in Figure 2), where the offset $\Delta\rho_{xx} = \rho_{xx}(T^*, H = 2T) - \rho_{xx}(T^*, H = 0)$ accounts for magnetoresistance of metallic phase at $H = 2T$. A magnetic field $H \gtrsim 500$ Oe is usually enough to destroy superconductivity in (TMTSF)$_2$ClO$_4$ [60,61], but we take the data at $H = 2T$ where the SC effects can be safely ignored. Since the experimental data on $\rho_{zz}(T)$ under magnetic field for the same samples are absent, the background-phase conductivity $\sigma_{zz}^b$ along the $z$-axis is found by extrapolating the metallic $\rho_{zz}(T)$ resistivity to low temperature by a second-order polynomial, similar to Ref. [61]. Here the second-order term comes from the electron-electron scattering at low temperature [61,62]. We take the $x$-axis as the reference axis for mapping to isotropic medium, i.e., $\sigma_{xx}^b$ is taken as the background isotropic conductivity in the mapped space: $\sigma^b = \sigma_{xx}^b$. According to Equation (A4), the mapping coefficient along the $z$-axis is defined as $\eta = \sigma_{zz}^b/\sigma_{xx}^b$.

The SC volume ratio $\phi$ is found from Equation (10) for $i = x$. The calculated volume ratio as a function of temperature is plotted in Figure 2. Substituting $\phi$ found from Equation (10) for $i = x$ to Equation (9) for $i = z$, we predict the effective resistivity along the $z$-axis, $\sigma_{zz}(T)$. Its comparison with the experimental data from Figure 2b of Ref. [6] is shown in Figure 3. We see a rather good agreement. In this calculation we have one fitting parameter—the unknown ratio of the semiaxes $a_x$ and $a_z$ of ellipsoidal SC inclusions. We found that at high temperature $T \approx 1.2$ K, typical inclusions have the aspect ratio $a_z/a_x \approx 0.16$. At low temperature $T \approx 0.5$ K, the inclusions have aspect ratio $a_z/a_x \approx 0.85$. It means that with a decrease in temperature the SC inclusion becomes more isotropic along x and z-axes, i.e., $a_z/a_x \to 1$. It may indicate the formation of large and almost isotropic clusters of small SC inclusions. Please note that at any temperature the found aspect ratio $a_z/a_x$ is much larger than the ratio of coherence lengths in (TMTSF)$_2$ClO$_4$, $\xi_z/\xi_x \approx 0.03$.

This supports the fact that in (TMTSF)$_2$ClO$_4$ the heterogeneity and SC islands originate from disorder and anion ordering rather than from usual SC fluctuations, because for SC fluctuations $a_z/a_x \sim \xi_z/\xi_x$.

Figure 3 and the inset in Figure 2 show the temperature dependence of resistivity for the same sample along the $z$ and $x$ axes correspondingly. From their comparison one observes that the resistivity drop near $T_c$ for $\rho_{zz}$ is much stronger than along for $\rho_{xx}$. This feature originates from the strong anisotropy of background-phase resistivity $\rho^b_{ii}$ and is naturally described in generalized MGA theory [32–34]. The qualitative interpretation of this anisotropic resistivity drop due to SC onset is illustrated in Figure 1 of Refs. [32] or [33]. Because of high resistivity $\rho^b_{zz}$, the interlayer current mainly flows via the SC islands serving as shortcuts for this current direction. The effective resistivity $\rho_{zz}$ is then determined by the much smaller intralayer resistivity and by the typical length of in-plane path between two close SC domains, which is inversely proportional to the SC volume fraction $\phi$.

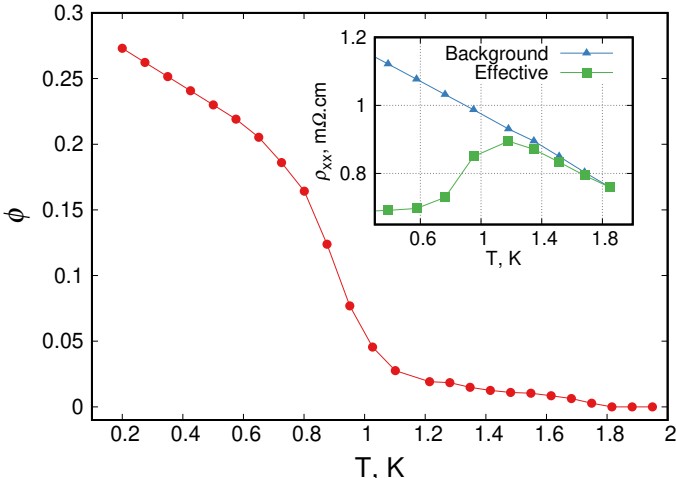

**Figure 2.** Dependence of SC volume ratio $\phi$ on temperature. $\phi$ (red,circle) is calculated using Equation (10) for $i = x$ and the experimental data on resistivity along the $x$-axis, taken from Figure 2c of Ref. [6] and shown in the inset. (inset) The effective medium resistivity $\rho_{xx}$ containing SC, metallic and SDW phases (green squares), and the background resistivity $\rho^b_{xx}$ in the absence of SC islands (blue triangles), taken from the data in Figure 2c of Ref. [6] in magnetic field.

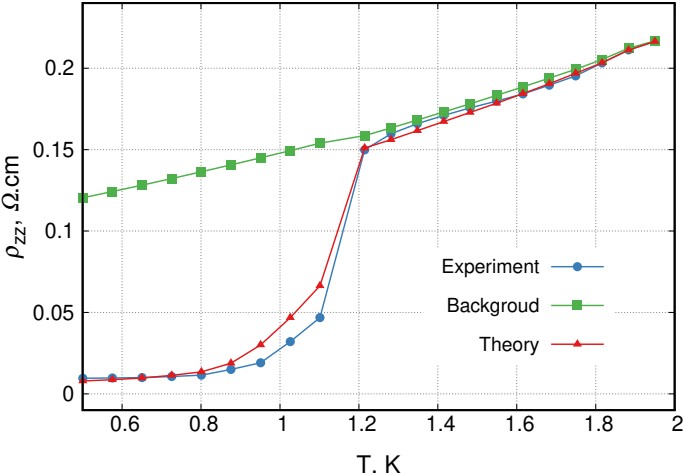

**Figure 3.** Temperature dependence of resistivity along $z$-axis. Experimental values (blue circles) are taken for sample #4 from Figure 2b of Ref. [6]. Background-phase metallic resistivity (green squares) is found by second-order approximation, i.e., $\rho^b_{zz} = 8.17 + 4.2\,T + 1.4\,T^2\,\Omega\cdot$cm. Theoretical values (red triangles) are found from $z$-axis resistivity in Equation (9).

### 4.2. Effect of Cooling Rate on Superconducting Volume Ratio

The cooling rate of $(TMTSF)_2ClO_4$ samples controls the fraction of $ClO_4$-ordered domains. At slow cooling, the $ClO_4$ anions have enough time to relax into ordered state. At fast cooling the thermal disorder remains in the samples, so that both ordered and disordered domains coexist. It was corroborated by resistivity [63], specific heat[64] and x-ray scattering [65] experiments.

The volume fraction $\phi_o$ of anion-ordered domains as a function of cooling rate has been studied using the X-ray scattering [65,66] and, recently, by resistivity and magnetic susceptibility [7] measurements. The corresponding results are compared in Figure 4 of Ref. [7] in the range of cooling rate $0 < -dT/dt < 20$ K/min. Several assumptions are made in extracting $\phi_o$ from these experiments[7,65]. First, (i) all these results assume that at the slowest cooling rate $-dT/dt|_{min} \sim 1$ K/min all $ClO_4$ are ordered at low temperature. It is not evident, as some degree of anion disorder may remain. Second, (ii) in the estimate of volume fraction $\phi_o$ from the resistivity measurements in the mixed SDW/metal phase in Ref. [7] the following equation (see Equation (1) of Ref. [7]) for the effective conductivity $\sigma_{zz}$ along $z$-axis has been used:

$$\sigma_{zz} = \left[ \phi \sigma_{min}^{1/3} + (1 - \phi) \sigma_{max}^{1/3} \right]^3, \tag{11}$$

where $\rho_{min} = 1/\sigma_{min} = 0.03$ $\Omega$·cm is taken as a residual resistance of the sample with lowest cooling rate and $\rho_{max} = 1/\sigma_{max} = \rho_{min} + \Delta\rho_{c*} = 0.26$ $\Omega$·cm is determined assuming that the difference $\Delta\rho_{c*} = \rho_{max} - \rho_{min} = 0.23$ $\Omega$·cm is equal to the jump of resistivity at anion-ordering temperature $T_{AO} = 24.5$ K due to the scattering by anion disorder. In fact, at low temperature the anion disorder has much stronger effect on conductivity than just the electron scattering by this disorder itself, because it also favors the formation of insulating SDW state. Even if the fraction of insulating SDW domains is about one half, as in Figure 4 of Ref. [7], it may considerably affect the electron conductivity. In addition, (iii) Equation (11) does not take into account the conductivity anisotropy of $(TMTSF)_2ClO_4$, which strongly enhances the effect of metal/SC domains on resistivity along the least-conducting axis [32–34], as given by Equation (8). (iv) The extraction of $\phi_o$ from the magnetic susceptibility $\chi(T)$ data, especially at rapid cooling rate when the SC volume fraction $\phi \ll 1$, depends strongly on the size and shape of SC domains [34,54]; therefore the assumption [7] that $\phi_o = [\chi(T \to 0) - \chi(T_c)]/[\chi(T \to 0) - \chi(T_c)]_{dT/dt=0.02K/min}$ is not valid when the size of SC domains is smaller than the London penetration depth.

In this subsection we estimate the volume fraction $\phi$ of SC phase in $(TMTSF)_2ClO_4$ using the resistivity data from Ref. [7] and applying Equations (9) and (10) derived in MGA. Please note that the volume fractions $\phi$ of SC phase and $\phi_o$ of anion-ordered phase may differ, e.g., because the former depends on temperature. Since the MGA approximation is valid only at $\phi \ll 1$, it can only give $\phi(T)$ at $T > T_c$. Thus, our method of estimating $\phi$ works better for higher cooling rate when $T_c$ is lower, therefore our results are rather complimentary to those in Ref. [7]. However, the cooling-rate dependence of $\phi(T > T_c)$ also gives the general tendency. Please note that at cooling rate $dT/dt = 100$ K/min and some annealing, by extrapolating the $\phi(T)$ curve in Figure 3 to $T = 0$ we obtain $\phi(T \to 0) \approx 0.3$, which is in a good agreement with other data in Figure 4 of Ref. [7].

To observe the influence of cooling rate on SC volume ratio $\phi(T)$, we use resistivity data from the inset of Figure 1d of Ref. [7]. These experimental values are taken as the effective resistivity $\rho_{zz}$ along $z$-axis for different cooling rates. For background-phase resistivity $\rho_{zz}^b$ along the $z$-axis we use the 2nd order polynomial fit of metallic resistivity at $T > T^*$. Substituting these $\rho_{zz}(T)$ and $\rho_{zz}^b$ in Equation (10) we obtain $\phi(T)$ for various cooling rates. Unfortunately, in Ref. [7] there is no resistivity data along other two axes which would allow us to find the ellipsoid aspect ratio. Therefore, in Figure 4 we take the depolarization factor $A_i = 1/3$, i.e., the spherical inclusions in the mapped space instead of ellipsoidal. This choice corresponds to ellipsoid semiaxes $a_i \propto \sqrt{\sigma_{ii}(T^*)} \propto \xi_i$ in real space, as one expects for SC fluctuations. (The metallic conductivity $\sigma_{ii} \propto v_i^2 \propto \xi_i^2$, where $v_i$ is the

Fermi velocity along axis $i$, and the BCS coherence length $\xi_i = \hbar v_i / \pi \Delta$.) The obtained $\phi(T)$ at cooling rates 0.02 K/min, 0.052 K/min, 2.5 K/min, 7.6 K/min and 18 K/min are shown in Figure 4. The curves in Figure 4 are similar to those in Figure 4 of Ref. [7], but the values of SC volume fraction $\phi$ are expectedly smaller than $\phi_o$ in Ref. [7], because at $T > T_c$ only a fraction of ClO$_4$-ordered domains becomes superconducting. However, $\phi$ increases with decreasing temperature and, probably, reaches $\phi_o$ at $T \to 0$.

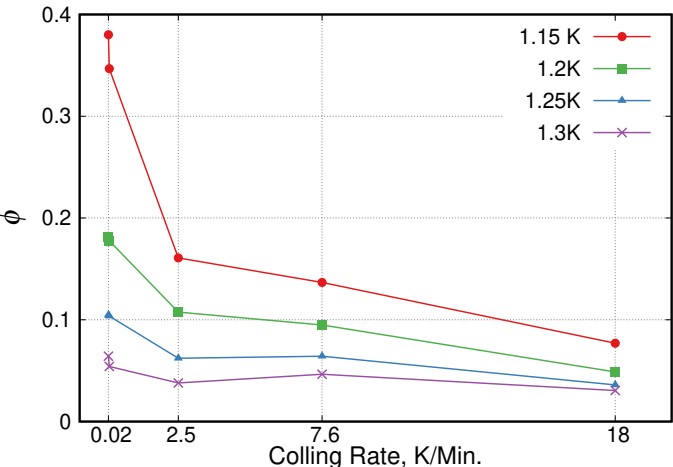

**Figure 4.** Dependence of SC volume ratio $\phi$ on cooling rate calculated using Equation (10) for different temperatures. For this calculation, the experimental data from the inset of Figure 1d in Ref. [7] are used.

*4.3. Effect of Disorder on the Shape of Superconducting Inclusions*

In Figure 4 we investigated the effect of cooling rate on $\phi$. However, along with $\phi$ the shape of SC domains also plays an important role. Recent work [31] has shown that the probability of percolation along the shortest sample dimension, i.e., along sample thickness, is higher than along other directions. It was corroborated by the experiment on FeSe [33,52], where, by reducing the $z$-axis thickness of sample from 300 nm to $\sim$50 nm, one raised $T_c$ from 8 K to 12 K [52]. The (TMTSF)$_2$ClO$_4$ samples are also usually flat. This effect of anisotropic superconductivity onset also depends on the shape of SC inclusions [31]. The knowledge of the shape of SC domains in (TMTSF)$_2$ClO$_4$ is also helpful to better understand the mechanism of their formation. In Section 4.1 we found $a_z/a_x$ for sample # 4 in Ref. [6] at cooling rate 100 K/min. Below we find $a_z/a_y$ and $a_y/a_x$ for the samples cooled at rate 600 K/min with various times of subsequent annealing. This gives the effect of disorder on the shape of SC inclusions.

To study the evolution of aspect ratios $a_z$:$a_y$:$a_x$ with disorder we use the experimental data from Figures 3 and 4 of Ref. [6]. Unfortunately, the curves with equal numbers in these two figures correspond to different samples. Thus, we do not have the data on resistivity along all three axes for the same sample and parameters, required to determine the full shape of ellipsoidal inclusions. However, we use the fact that the depolarization factors $A_i$ in Equation (4) depend most strongly on the semiaxis $a_i$ along the same direction, which allows us to vary only one parameter for each fit.

First we find the evolution of $a_z/a_y$ with disorder. From the resistivity data along the $z$-axis, given in Figure 4a of Ref. [6], using Equation (10) we find $\phi(T)$ for different degrees of disorder. Using these $\phi(T)$ in Equation (9) we predicted resistivity along $y$-axis. From best fit values of predicted and experimental resistivity along $y$-axis, given in Figure 4b of Ref. [6], we find the ratio $a_z/a_y$ for various degrees of disorder. The results are shown in Figure 5a and reveal that at low disorder, up to sample #9, the ratio $a_z/a_y$ almost remains same. However, at higher disorder the ratio $a_z/a_y$ increases.

Similarly, to find the evolution of $a_y/a_x$ with disorder we use the data from Figure 3 of Ref. [6]. As before, from the resistivity data along $y$-axis using Equation (10) we find

$\phi(T)$ for different degrees of disorder. Using this $\phi(T)$ we predict resistivity along $x$-axis. We change the semiaxes of ellipsoids along $y$- and $x$-direction, so that the theoretical and experimental values of resistivity agree. Thus, the obtained ratio of $a_y/a_x$ is shown in Figure 5b.

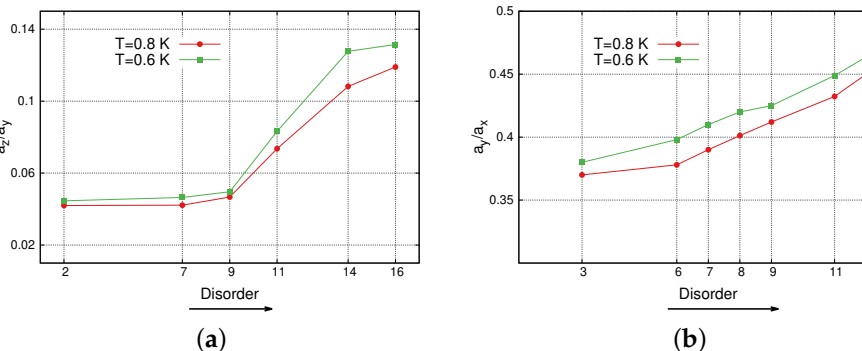

**(a)**　　　　　　　　　　　　　　　　　　　　**(b)**

**Figure 5.** The dependence of aspect ratios $a_z/a_y$ (**a**) and $a_y/a_x$ (**b**) of superconducting domains in (TMTSF)$_2$ClO$_4$ on disorder at two temperatures $T = 0.8$ K and 0.6 K, calculated using Equations (4), (9) and (10) and resistivity data from Figures 3 and 4 of Ref. [6], taken at cooling rate 600 K/min and various annealing times. At longer annealing time, i.e., at weaker disorder, the shape of SC domains is more anisotropic.

## 5. Discussion

In this paper, we propose a method based on MGA to investigate the microscopic parameters of heterogeneous superconductors from resistivity data. We apply this method to the organic superconductor (TMTSF)$_2$ClO$_4$, where SC coexists with SDW in the form of isolated domains. Using our method we study the SC volume fraction $\phi$ and the shape of SC islands as a function of external parameters, such as temperature and ClO$_4$ anion disorder, which can be experimentally controlled by the cooling rate through the anion-ordering transition at $T_{AO} \approx 24.5$ K [7,65] or by the annealing of (TMTSF)$_2$ClO$_4$ samples [6].

For the best use of proposed method, one needs the following experimental data: (i) Temperature dependence of resistivity $\rho_{ii}(T)$ along each of non-equivalent main crystal axes (If the crystal has orthorhombic or lower symmetry, one needs the data along all three axes. If two or three crystal main axes are equivalent by symmetry, one only needs the data along two or one axes correspondingly.); (ii) $\rho_{ii}(T, H_0)$ in a magnetic field $H_0 > H_c$ destroying SC, to get the resistivity $\rho_{ii}^b(T)$ of the background homogeneous phase. In the absence of $\rho_{ii}(T, H_0)$ or in the case of non-SC inclusions, one needs to make an extrapolation of $\rho_{ii}(T)$ from $T > T^*$ to lower $T$ to get $\rho_{ii}^b(T)$, which is less accurate. In the case of SDW/CDW inclusions one can also apply an external pressure destroying SDW/CDW to get $\rho_{ii}^b(T)$. If also (iii) magnetic susceptibility data are present, especially for all non-equivalent magnetic-field orientations, they help to independently check the obtained microscopic parameters and allow the estimate of the average size of SC inclusions as compared to the SC penetration depth [33,34]. Ideally, all these data are available for several values of external parameters that one is interested in, for example, at each studied cooling rate of (TMTSF)$_2$ClO$_4$. Unfortunately, despite an active experimental investigation of (TMTSF)$_2$ClO$_4$ by resistivity measurements, e.g., performed recently in Refs. [6,7], this full set of data is absent. Nevertheless, we have analyzed the available data from Refs. [6,7] to make some physical predictions concerning the mixed SC/SDW phase in (TMTSF)$_2$ClO$_4$.

In Figures 2 and 4 we show the obtained SC volume fraction $\phi$ for various cooling rates and temperatures $T_c < T < T^*$. Figure 3 illustrates how well the MGA model typically fits the experimental data. Figures 5 and 6 show the evolution of the shape of SC grains with the change of disorder by the annealing of rapidly cooled samples.

From Figure 5a we observe that for partially ordered samples #2, #7, #9 the aspect ratio $a_z/a_y \approx 0.05$ of SC domains at temperature $T \approx 0.6$–0.8 K depends weakly on disorder. The corresponding SC volume ratio $\phi \approx 0.1$ also weakly depends on disorder. At shorter

annealing time, i.e., at larger disorder, the SC volume ratio decreases to $\phi \approx 0.03$, while the aspect ratio increases to $a_z/a_y \approx 0.13$. Another aspect ratio $a_y/a_x \approx 0.4$ depends much weaker on disorder, as shown in Figure 5b. Please note that the obtained ratios $a_x:a_y:a_z$ at cooling rate 600 K/min are close to those of SC coherence length. In Figure 2 we found that for partially ordered sample #4 at cooling rate 100 K/min the aspect ratio of SC domains $a_z/a_x \approx 0.85$ at temperature $T \approx 0.5$ K, which differs considerably from what we get at 600 K/min even for long annealing time. This means that the decrease of cooling rate, as in Refs. [7,65] is not completely equivalent to the increase of annealing time at $T < T_{AO}$ used in Ref. [6]: they have similar effect on SC volume fraction $\phi$ but different effect on the shape of SC domains. Therefore, it would be very interesting to study their effect on the SC domain size, which can be extracted from the simultaneous magnetic susceptibility measurements as done for other compounds [33,34].

In organic superconductors $(TMTSF)_2ClO_4$ [6] and $(TMTSF)_2PF_6$ [14,15] superconductivity onsets anisotropically, i.e., first along the highest conducting $z$-axis and only in the end along the lowest-conducting $x$-axis. This behavior was first explained by assuming filamentary SC inclusions elongated along $z$-axis [14], but this hypothesis received neither theoretical nor experimental proof till now. Our analysis also shows that the SC domains are not elongated along $z$-axis but, on contrary, are oblate. Nevertheless, we predict much stronger decrease of resistivity along the least-conducting direction (compare Figure 3 and the inset in Figure 2), similar to experimental observations in $(TMTSF)_2ClO_4$ [6] and in many other heterogeneous anisotropic superconductors [32–34]. In our recent work [31] we proposed a simple model to explain the anisotropic zero-resistance onset also. We have shown [31] that the percolation probability along SC islands in needle or flat shaped samples is the highest along the shortest direction. A schematic illustration of this idea is given in Figure 6. The same idea can be applied to $(TMTSF)_2ClO_4$ to explain the anisotropic onset of superconductivity. Usually, the $(TMTSF)_2ClO_4$ samples are much shorter along the interlayer $z$-axis than along other two, e.g., the dimensions of samples in Ref. [6] are $3 \times 0.1 \times 0.03$ mm$^3$. Hence, the probability of percolation for the ellipsoidal inclusion, even with the obtained anisotropic aspect ratios $a_z/a_x$ and $a_z/a_y$, will be the highest along the $z$-axis. Our preliminary calculations of percolation threshold for such flat elongated samples confirm this statement and show that the zero resistance, i.e., the percolation threshold, can be achieved even when the SC volume ratio $\phi = \phi_c \ll 1$. Hence, without invoking a filamentary SC one can easily explain the anisotropic onset of superconductivity in organic metals.

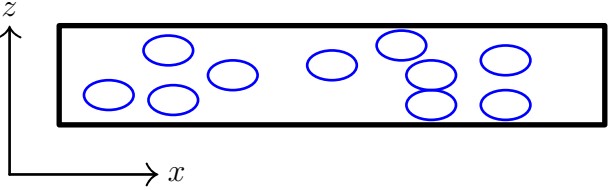

**Figure 6.** Schematic illustration of superconducting ellipsoid inclusions embedded inside a long thin conductor with dimensions $L_z \ll L_x$. It shows that the percolation probability along $z$ is higher than along $x$-axis if $a_z/a_x > L_z/L_x$.

In our semi-classical theory based on MGA we neglect the dynamic fluctuations. Probably, the position and size of domains may fluctuate. However, in the studied organic metal the competition between SDW and superconductivity, leading to the domain formation, is governed by the disorder of $ClO_4$ anion orientation, which is frozen well below the anion-ordering transition temperature $T_{AO} \approx 24.5$ K. Hence, below $T_{SDW} \approx 4$–5 K we may consider the metal/SC and SDW domains as time-independent.

For all our calculation in Section 4 we have used Equations (9) and (10) because the inclusions are superconducting. However, the similar approach and Equation (8) can be used when the conductivity of inclusions is finite, e.g., when SDW inclusions are embedded inside a metallic background or vice versa. This is a usual occurrence in organic supercon-

ductors [4]. Thus, instead of using Equation (11) or similar phenomenological formulas to analyze the resistivity data in mixed metal/SDW or metal/CDW phases in organic metals, we recommend using MGA Formulas (8) and (4), which take into account the strong anisotropy of layered organic metals and the actual shape of domains. Apart from that, in several high-Tc superconductors as $Bi_2Sr_xCa_{3-x}Cu_2O_8$ [67], $La_{2-x}Sr_xCuO_4$ [68], $Ba(Fe_{1-x}Co_x)_2As_2$ [69–71], and in many other materials, e.g., $HfTe_3$ [72], there is also a spatially separated coexistence of metallic and SC or SDW/CDW phases. Hence in all these heterogeneous materials, provided the conductivity of both metallic and SDW/CDW phases is known and the domain size exceeds the coherence length, one can estimate the second-phase volume ratio and the domain shape from resistivity data using the above method.

## 6. Conclusions

Using the Maxwell-Garnett approximation we estimate the volume fraction and the shape of superconducting domains from resistivity measurements. We apply this method to investigate the heterogeneous electronic structure in organic superconductor $(TMTSF)_2ClO_4$, where superconductivity coexists with the spin-density wave in the form of isolated domains. This material is especially important to study such coexistence because it appears even at ambient pressure and can be easily controlled by changing the cooling rate or annealing time of the samples. From available resistivity data we study the evolution of the volume fraction and of the shape of superconducting domains in $(TMTSF)_2ClO_4$ with disorder and temperature. Our method applies not only to superconductors, but also when the density-wave or other-type domains are embedded inside a metallic background, or vice versa.

**Author Contributions:** P.D.G. conceptualized and developed the Methodology. Formal analysis, validation, manuscript writing and review was done by K.K.K., P.D.G. and V.D.K. All authors have read and agreed to the published version of the manuscript.

**Funding:** This article is partly supported by the Ministry of Science and Higher Education of the Russian Federation in the framework of Increase Competitiveness Program of MISiS, by RFBR grant No. 21-52-12027, and by the "Basis" Foundation for development of theoretical physics and mathematics. V. D. K. acknowledges the MISiS project No. K2-2020-001, and K. K. K. the MISiS support project for young research engineers and RFBR grants Nos. 19-32-90241 & 19-31-27001. P. D. G. acknowledges the State Assignment No. 0033-2019-0001 and RFBR grants Nos. 19-02-01000 & 21-52-12043.

**Institutional Review Board Statement:** Not applicable.

**Informed Consent Statement:** Not applicable.

**Data Availability Statement:** Data available on request.

**Conflicts of Interest:** The authors declare no conflict of interest.

## Appendix A. Anisotropic Dilation of the Problem of Static Current Distribution

Anisotropic medium is converted to isotropic one by mapping the real space to a mapped space, where the solution is simpler. The mapping should satisfy the following conditions: (i) the conductivity of background phase in the mapped space should be isotropic, and (ii) the electrostatic continuity equation should be satisfied in the mapped space with the same solution.

In our notations, $\sigma_{xx}^b, \sigma_{yy}^b, \sigma_{zz}^b$ are the constant conductivity components of background phase in the original heterogeneous medium. Let $J$ and $V$ be the current density and the applied potential, respectively. The electrostatic continuity equation in the background phase is then written as

$$-\nabla J = \sigma_{xx}^b \frac{\partial^2 V}{\partial x^2} + \sigma_{yy}^b \frac{\partial^2 V}{\partial y^2} + \sigma_{zz}^b \frac{\partial^2 V}{\partial z^2} = 0. \tag{A1}$$

Heterogeneity is hidden in the boundary conditions on the surface of each grain and of the sample.

Let $x'$, $y'$ and $z'$ be the axes in mapped space, where conductivity should be isotropic: $\sigma^b_{x'x'} = \sigma^b_{y'y'} = \sigma^b_{z'z'} = \sigma^b$. If $J'$ and $V'$ are the current density and electrostatic potential respectively in the mapped space, the continuity equation in the mapped space is written as

$$-\nabla' J' = \sigma^b \left( \frac{\partial^2 V'}{\partial x'^2} + \frac{\partial^2 V'}{\partial y'^2} + \frac{\partial^2 V'}{\partial z'^2} \right) = 0. \tag{A2}$$

The condition (ii) for our mapping means that the solution

$$V(x, y, z) = V'(x', y', z') \tag{A3}$$

of Equations (A1) and (A2) is the same. This solution completely determines the effective conductivity of heterogeneous medium, as it also gives the current density via $J_i = \sigma^b_{ii} \nabla_i V \neq J'_i = \sigma^b \nabla'_i V'$. Equations (A1)–(A3) are consistent if the mapping is the anisotropic scaling, e.g.,

$$x = x', \qquad y = \sqrt{\mu} y', \qquad z = \sqrt{\eta} z', \tag{A4}$$

with constant mapping coefficients $\mu$ and $\eta$ determined by conductivity anisotropy in real space:

$$\mu = \frac{\sigma^b_{yy}}{\sigma^b_{xx}}, \qquad \eta = \frac{\sigma^b_{zz}}{\sigma^b_{xx}}. \tag{A5}$$

Instead of (A4) one could choose a product of the mapping (A4) and of any isotropic scaling by a factor $\alpha$ with the simultaneous change of $\sigma^b \to \alpha^2 \sigma^b$. We have chosen $\sigma^b = \sigma^b_{xx}$, so that $\mu, \nu < 1$.

The current components in the real space $r = (x, y, z)$ and in the mapped space $r' = (x', y', z')$ are related as

$$J_x(r) = J'_{x'}(r'), \qquad J_y(r) = \sqrt{\mu} J'_{y'}(r'), \qquad J_z(r) = \sqrt{\eta} J'_{z'}(r'). \tag{A6}$$

The shapes of inclusions are not preserved during this mapping procedure. For example, a sphere with radius $a_x$ described by the equation $x^2/a_x^2 + y^2/a_x^2 + z^2/a_x^2 = 1$ in non-homogeneous medium will transform to an ellipsoid described by the equation $x'^2/a_{x'}^2 + y'^2/a_{y'}^2 + z'^2/a_{z'}^2 = 1$ in mapped space, where the semiaxes are given by

$$a_{x'} = a_x, \quad a_{y'} = a_x/\sqrt{\mu}, \quad a_{z'} = a_x/\sqrt{\eta}. \tag{A7}$$

If $z$ is the lowest-conducting axis, and the highest conducting axis is $x$, then a sphere in real space transforms to an ellipsoid elongated along $z$-axis. Due to the temperature dependence of conductivity anisotropy, the coefficients $\mu, \eta$ and the shape of inclusions change with temperature either. If in real space the inclusion is ellipsoid with semiaxes $a = a_x$, $b = \beta a_x$ and $c = \gamma a_x$, then it transforms to ellipsoid in mapped space with semiaxes

$$a_{x'} = a_x, \quad a_{y'} = a_x \beta/\sqrt{\mu}, \quad a_{z'} = a_x \gamma/\sqrt{\eta}. \tag{A8}$$

In MGA we take ellipsoidal inclusions with fixed aspect ratios $\beta$ and $\gamma$, but varying size.

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
