# Peer review of "Evolution of Shape and Volume Fraction of Superconducting Domains with Temperature and Anion Disorder in (TMTSF)2ClO4"

_crystals, doi:10.3390/cryst11010072_

Round 1

Reviewer 1 Report

In this paper, the authors report Maxwell-Garnett approximation analyses to explain the cooling rate dependence of the resistivity of (TMTSF)2ClO4 produced by the appearance of the superconducting region embedded in the metallic and SDW phase. They claim that phenomenological calculations based on this model can explain the small volume fraction and variation of it depending on the cooling rate. They included disorder effects and discuss that the anisotropy also changes with the annealing. It seems informative for understanding the nature of this competitive nature in the electronic phases. I have several comments as follows.

  1. The definition of various transition temperatures are defined in the section 2.1. However, the authors mention T*, probably the onset-temperature of superconductivity without definition in this section.
  2. The SC percolation threshold fc in the same section should be explained more clearly. How do the authors evaluate this value in this theory?
  3. Since the microscopic competition of metallic region and superconductive region should be percolative and emerges with dynamic fluctuations, just the arrangement of the microdomain as is shown for example in Figure 6 do not include such dynamic features. Probably, fluctuations of the position of the domains and also those in size may exist. I think it is better to explain how did the authors treat such dynamic effects in this semi-classical theory?
  4. The authors calculated the zero-resistivity mainly based on current distributions in the sample assuming that the MGA model is held. The cooling rate dependence appears in the volume of the Meissner fraction and the heat capacity peak. If it is possible, it seems better to mentioned on the volume fraction appears in such thermodynamic properties.

I think this paper can be considered for publication, if the authors include these points in appropriate form.

Author Response

We thank the reviewer for reading our manuscript and sending his/her comments. We corrected the manuscript according to these comments (items 4-6 in the list of changes). Below is our reply to the referee comments/questions.

  1. In the section 2.1 T* is now defined: “Here T* is the superconducting onset temperature.”
  2. We refer to our previous paper [1] for the calculation of percolation threshold fc, where this calculation is described in detail. It is calculated numerically, assuming rectangular sample and ellipsoidal SC inclusions of various size with randomly distributed but fixed positions. Within MGA theory the percolation threshold cannot be calculated, as MGA always gives fc=1. To make this point clearer in the manuscript, in the beginning of Sec. 2.2 we replaced the first two sentences by a new paragraph.
  3. In our semi-classical theory we neglect the dynamic fluctuations. Probably, the fluctuations of position of the domains and also those in size may exist. But in the studied organic metal the competition between SDW and superconductivity, leading to domain formation, is governed by the disorder of ClO4 anion orientation, which is frozen below the anion-ordering transition temperature. Hence, we may consider the metal/SC and SDW domains as time independent.
  4. We calculated the zero-resistivity not using the MGA model, but using the numerical calculations of percolation threshold. The MGA is used only to describe the temperature dependence of resistivity above Tc. It is true that the cooling-rate dependence appears in the volume of the Meissner fraction and the heat capacity peak. Unfortunately, the corresponding experimental data in Ref. [8] by S.Yonezawa et al. are present only at slow cooling rate <18K/min and cannot be directly compared (using our model) with the results of Ref. [7] by Ya.Gerasimenko et al. Moreover, as we noted in our paper on page 9, the extraction of $\phi $ from the magnetic susceptibility $\chi (T)$ data, especially at rapid cooling rate when the SC volume fraction $\phi << 1$, depends strongly on the size (compared to SC penetration depth) and shape of SC domains.

********************

List of changes, made according to referee comments.

1) The article title is changed. The first sentence of Conclusions section is modified.

2) In section 3 the variables are defined.

3) Misprints are corrected. For example, in line 179 on page 7 the extra phrase "to describe resistivity" is removed.

4) In the section 2.1 T* is defined: “Here T* is the superconducting onset temperature.”

5) In the beginning of Sec. 2.2 we rewrote the first 2 sentences and extended them to the whole first paragraph of Sec. 2.2, describing the calculation method and results on percolation threshold fc, performed in Ref. [1] and relevant to the range of applicability of MGA approximation used in our paper.

6) A last but one paragraph is added in the Discussion section:

“In our semi-classical theory based on MGA we neglect the dynamic fluctuations. Probably, the position and size of domains may fluctuate. But in the studied organic metal the competition between SDW and superconductivity, leading to the domain formation, is governed by the disorder of ClO4 anion orientation, which is frozen well below the anion-ordering transition temperature $T_{AO} \approx 24.5\:K$. Hence, below $T_{SDW} \approx 4-5 \: K$ we may consider the metal/SC and SDW domains as time independent.”

Reviewer 2 Report

The authors report a method to evaluate the volume fraction of superconducting domains in materials, and discuss the evolution of superconducting domains in an organic superconductor (TMTSF)2ClO4.   The good agreement with the experimental data for ρzz demonstrates validity of the analysis and other results for the volume fraction and the domain size are well discussed, and I think the data deserve publication.

In some places, especially in title, the author wrote as if one of the main outcomes of the paper is the development of the method.  However, the novelty of the methodology is not clear.  The analysis described in the manuscript is very similar to that reported in the authors’ previous paper (ref.34).  I think the author should modify the manuscript, including the title, so as to show their new outcomes properly.

As a minor points,

- In section 3, many variables are not defined.

- In line 179 on page 7, "to describe resistivity" should be removed.

Author Response

We thank the reviewer for reading our manuscript and sending his/her comments. We corrected the manuscript according to these comments (points 1-3 in the list of changes).

********************

List of changes, made according to referee comments.

1) The article title is changed. The first sentence of Conclusions section is modified.

2) In section 3 the variables are defined.

3) Misprints are corrected. For example, in line 179 on page 7 the extra phrase "to describe resistivity" is removed.

4) In the section 2.1 T* is defined: “Here T* is the superconducting onset temperature.”

5) In the beginning of Sec. 2.2 we rewrote the first 2 sentences and extended them to the whole first paragraph of Sec. 2.2, describing the calculation method and results on percolation threshold fc, performed in Ref. [1] and relevant to the range of applicability of MGA approximation used in our paper.

6) A last but one paragraph is added in the Discussion section:

“In our semi-classical theory based on MGA we neglect the dynamic fluctuations. Probably, the position and size of domains may fluctuate. But in the studied organic metal the competition between SDW and superconductivity, leading to the domain formation, is governed by the disorder of ClO4 anion orientation, which is frozen well below the anion-ordering transition temperature $T_{AO} \approx 24.5\:K$. Hence, below $T_{SDW} \approx 4-5 \: K$ we may consider the metal/SC and SDW domains as time independent.”